# Modulation of RNA processing genes during sleep-dependent memory

Yongjun Li[1,2†], Nitin S Chouhan[1†‡], Shirley L Zhang[1§], Rebecca S Moore[1], Sara B Noya[1], Joy Shon[1], Zhifeng Yue[1], Amita Sehgal[1]*

[1]Howard Hughes Medical Institute and Chronobiology and Sleep Institute, Perelman School of Medicine at the University of Pennsylvania, Philadelphia, United States; [2]Department of Biology, University of Pennsylvania, Philadelphia, United States

**\*For correspondence:**
amita@pennmedicine.upenn.edu

[†]These authors contributed equally to this work

**Present address:** [‡]Department of Biological Sciences, Tata Institute of Fundamental Research, Mumbai, India; [§]Department of Cell Biology, Emory University School of Medicine, Atlanta, United States

**Competing interest:** The authors declare that no competing interests exist.

## eLife Assessment

The aim of this **important** study is to identify novel genes involved in sleep regulation and memory consolidation. It combines transcriptomic approaches following memory induction with measurements of sleep and memory to discover molecular pathways underlying these interlinked behaviors. The authors explore transcriptional changes in specific mushroom body neurons and suggest roles for two genes involved in RNA processing, Polr1F and Regnase-1, in the regulation of sleep and memory. Their findings offer **convincing** evidence that the expression of RNA processing genes is modulated during sleep-dependent memory, with Polr1F potentially contributing to increased sleep.

**Abstract** Memory consolidation in *Drosophila* can be sleep-dependent or sleep-independent, depending on the availability of food. The anterior posterior (ap) alpha'/beta' (α'/β') neurons of the mushroom body (MB) are required for sleep-dependent memory consolidation in flies fed after training. These neurons are also involved in the increase of sleep after training, suggesting a coupling of sleep and memory. To better understand the mechanisms underlying sleep and memory consolidation initiation, we analyzed the transcriptome of ap α'/β' neurons 1 hr after appetitive memory conditioning. A small number of genes, enriched in RNA processing functions, were differentially expressed in flies fed after training relative to trained and starved flies or untrained flies. Knockdown of each of these differentially expressed genes in the ap α'/β' neurons revealed notable sleep phenotypes for Polr1F and Regnase-1, both of which decrease in expression after conditioning. Knockdown of Polr1F, a regulator of ribosome RNA transcription, in adult flies promotes sleep and increases pre-ribosome RNA expression as well as overall translation, supporting a function for Polr1F downregulation in sleep-dependent memory. Conversely, while constitutive knockdown of Regnase-1, an mRNA decay protein localized to the ribosome, reduces sleep, adult specific knockdown suggests that effects of Regnase-1 on sleep are developmental in nature. We further tested the role of each gene in memory consolidation. Knockdown of Polr1F does not affect memory, which may be expected from its downregulation during memory consolidation. Regnase-1 knockdown in ap α'/β' neurons impairs all memory, including short-term, implicating Regnase-1 in memory, but leaving open the question of why it is downregulated during sleep-dependent memory. Overall, our findings demonstrate that the expression of RNA processing genes is modulated during sleep-dependent memory and, in the case of Polr1F, this modulation likely contributes to increased sleep.

## Introduction

Sleep is an optimized state for memory consolidation compared to wake (*Rasch and Born, 2013*). Indeed, sleep is thought to reorganize and strengthen the neural connections required for novel memory formation and long-term memory consolidation, and sleep disruption leads to impaired memory consolidation (*Roselli et al., 2021*). Beneficial effects of sleep on memory have also been attributed to post-learning neuronal reactivation (*Wagner et al., 2007*; *Dag et al., 2019*).

We recently demonstrated that *Drosophila* switch between sleep-dependent and sleep-independent memory consolidation based on food availability (*Chouhan et al., 2021*). Flies that are fed after appetitive conditioning show sleep-dependent memory consolidation, which is mediated by the anterior posterior (ap) α'/β' neurons of the mushroom body. On the other hand, flies starved after training display sleep-independent memory mediated by medial (m) neurons of the α'/β' lobes. Neurotransmission from ap α'/β' neurons in the first 4 hr after training in the fed flies is not only required for long-term memory but also increased sleep post-training. Indeed, activation of ap α'/β' neurons promotes baseline sleep, supporting the idea that post training sleep is triggered by the same neurons that are required for sleep-dependent memory consolidation. However, how sleep and memory consolidation are coordinated in ap α'/β' neurons in this time window is not understood.

Since both post-training sleep and memory consolidation occur rapidly within a few hours after training, we investigated transient gene expression changes in the ap α'/β' neurons after training to elucidate the interplay of sleep and memory consolidation. Here, we profiled the transcriptome of ap α'/β' neurons 1 hr after flies were fed and trained, as well as under control conditions in which flies were starved and trained or fed and untrained. We knocked down the differentially expressed genes in trained and fed flies and identified two RNA processing genes that affect sleep. Knockdown of one of these, Polr1F, a regulator of ribosomal RNA synthesis, promotes sleep and translation, both of which are required for consolidation of memory in trained and fed flies. Knockdown of Polr1F does not affect memory, which is consistent with downregulation of this gene during memory consolidation. Knockdown of Regnase-1, an mRNA decay protein, reduces sleep and disrupts post-training sleep, and also impairs both short- and long-term memory. While the relevance of Regnase-1 downregulation during sleep-dependent memory is unclear, overall we propose that RNA processing is modulated during sleep-dependent memory, likely to regulate sleep and memory.

## Results

### Transient transcriptome profiling of ap α'/β' neurons after training

We previously demonstrated that flies fed after appetitive memory training exhibit increased sleep and form sleep-dependent memory, which is mediated by ap α'/β' neurons of the mushroom body (MB) α'/β' lobes (*Chouhan et al., 2021*). To address the mechanisms that mediate increases in sleep and consolidation of memory in ap α'/β' neurons, we assayed gene expression changes in ap α'/β' neurons of flies fed after training in an appetitive conditioning paradigm. We crossed ap α'/β' neuron driver *R35B12-Gal4* (BDSC #49822) with *UAS-nGFP* (BDSC #4775) to label all ap α'/β' neurons with nuclear GFP and collected 5- to 7-day-old F1 progeny, which were subjected to one of three different conditions: Trained-Fed, Trained-Starved, Untrained-Fed as illustrated (*Figure 1A*). The trained-starved flies served as controls for sleep-dependent changes, while flies that were fed but untrained served as controls for training-dependent changes. After 1-hr, 50 mixed sex (25 for each sex) fly brains from each condition were dissected, and 500 GFP+ cells were sorted for bulk-RNA sequencing using the protocol described by Hongjie *Li et al., 2017*.

Our analysis of the bulk RNA-seq data revealed that most genes are not altered in ap α'/β' neurons after one hour, so there are no significant global changes observed among the three conditions. The correlation matrix calculated using the top 75% genes showed high similarity between samples, with most values exceeding 0.9 (*Figure 1—source data 1*). Nonetheless, we did observe a small subset of genes that were rapidly responsive and differentially expressed between the groups (*Figure 1B*). Pathway Principal Component Analysis (PathwayPCA) of these differentially expressed genes (DEGs), a statistical method to identify key patterns in gene expression data by reducing data dimensionality, showed that genes contributing to PC1, which accounted for 35% of the variance, largely encode proteins involved in transcription and biosynthesis, including RNA biosynthesis processes (*Figure 1—figure supplement 1A*; *Odom et al., 2020*). This suggests that the training and feeding paradigm

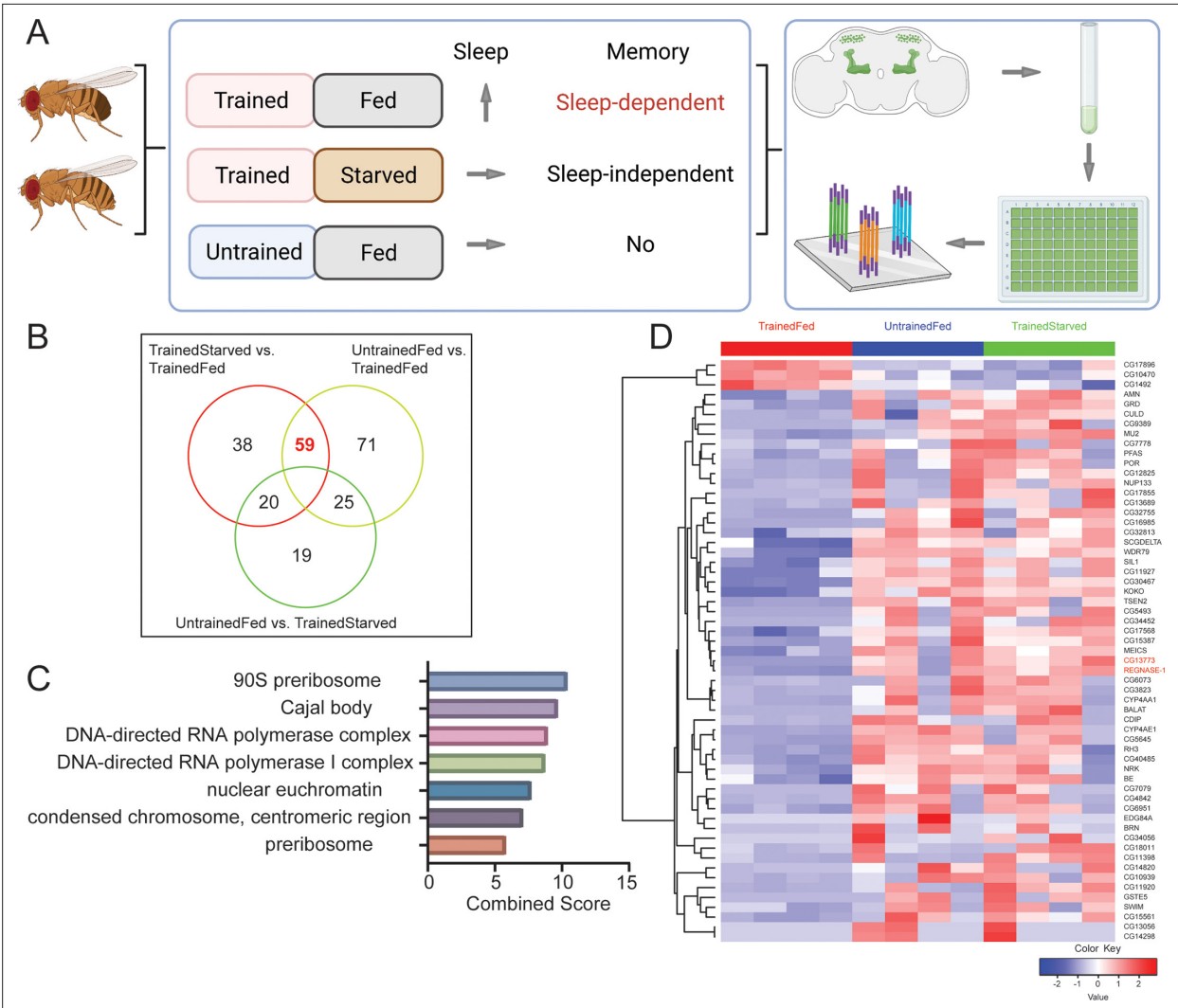

**Figure 1.** Differential gene expression after training in mushroom ap α′/β′ neurons. (**A**) 5- to 7-day-old mixed sex *wCS* flies were exposed to one of the following three conditions: Trained-Fed, Trained-Starved and Untrained-Fed. Only Trained-Fed flies are expected to increase sleep after treatment and thus form sleep-dependent memory (*Chouhan et al., 2021*). Brain dissection, single cell suspension and cell sorting were used to extract ap α′/β′ neurons in each of these three different conditions, and bulk-sequencing of the sorted cells was conducted (created in BioRender.com/k87y777). (**B**) We sequenced four samples for each condition and subjected them to differential gene analysis of pairwise comparison of the three conditions. We found that 59 genes are significantly different in TrainedStarved vs. TrainedFed and UntrainedFed vs. TrainedFed. (**C**) Gene ontology (GO) analysis of these 59 genes using FlyEnrichr (https://maayanlab.cloud/FlyEnrichr/) revealed that they encode cellular components of the 90 S preribosome, Cajal body, DNA-directed RNA polymerase complex, nuclear euchromatin, and condensed chromosome.(**D**) Heatmap of the 59 DEGs including two genes labeled in red, CG13773 (Polr1F) and Regnase-1, which affect sleep and are the focus of this study due to their impacts on sleep. DEGs were identified by DESeq2 with the cutoff of FDR <0.1 and fold change >1.5. Heatmaps were plotted by using TPM values of genes for each sample; data were log-transformed and scaled row-wise for visualization.

The online version of this article includes the following source data and figure supplement(s) for figure 1:

**Source data 1.** RNA-sequencing samples and differentially expressed genes information.

**Figure supplement 1.** PathwayPCA visualization and GO analysis of differentially expressed genes across three conditions.

influenced transcription and RNA biosynthetic processes. Meanwhile, gene ontology analysis of these DEGs, performed using DAVID Bioinformatics (https://david.ncifcrf.gov/home.jsp), revealed significant enrichment in processes such as protein unfolding/refolding and the stress response (*Figure 1—figure supplement 1B*; *Sherman et al., 2022*).

Of all the DEGs across the three groups, we focused on 59 DEGs that were significantly different in TrainedStarved vs. TrainedFed and UntrainedFed vs. TrainedFed; of these, 56 were downregulated

and only three were upregulated in the Trained-Fed condition compared to the control conditions (*Figure 1B, D*, *Figure 1—source data 1*). Gene ontology (GO) analysis of these 59 genes using FlyEnrichr (https://maayanlab.cloud/FlyEnrichr/) indicated that they encode cellular components of the 90 S preribosome, Cajal body, DNA-directed RNA polymerase complex, nuclear euchromatin, and condensed chromosome, consistent with the PCA enrichment (*Figure 1C*; *Chen et al., 2013*). Our transcriptome results suggest that ribosome biosynthesis and transcription are the initial changes in ap α′/β′ neurons of trained and fed flies.

## Two genes expressed differentially in ap α′/β′ neurons of trained and fed flies predominately affect sleep

To investigate if any of the 59 DEGs identified in the ap α′/β′ neurons of trained and fed flies affect baseline sleep, we knocked down each of the genes using UAS-RNAi lines and screened for their potential effects on sleep. We used the ap α′/β′ neuron constitutive driver R35B12-Gal4 line to drive the RNAi constructs and compared the sleep patterns of knockdown flies with those of R35B12-Gal4 and UAS-RNAi control flies (*Figure 2A and B*, *Figure 2—source data 1*). Using a cutoff of a 200 min change in sleep relative to each control group, we identified two genes, Polr1F and Regnase-1, that showed significant effects on baseline sleep. Knockdown of Polr1F, a component of RNA polymerase I complex, led to an increase in sleep. Although Polr1F has not been extensively studied in *Drosophila* (*Marygold et al., 2020*), its human ortholog hRPA43 is part of the multi-subunit protein complex Pol I that regulates the transcription of ribosomal RNA (*Beckouët et al., 2011*). Knockdown of Regnase-1, an RNA-binding protein that binds to mRNA undergoing active translation and promotes mRNA decay via its ribonuclease activity, in ap α′/β′ neurons resulted in a reduction of both nighttime and daytime sleep (*Figure 2C–F*). Further analysis of sleep architecture revealed that knockdown of Polr1F and Regnase-1 did not significantly impact the total activity of flies, while knockdown of either Polr1F and Regnase-1 resulted in increased and decreased average length of sleep episodes, respectively (*Figure 2—figure supplement 1A-F*).

Given that these two genes reduce expression rapidly in response to training under fed conditions, we next used the inducible pan-neuronal GeneSwitch driver *nSyb-GeneSwitch (GS)* to determine if restricting knockdown of these two genes to the adult stage recapitulates changes in sleep and/or memory. RU486 (mifepristone) was added to normal fly food and transgene expression is induced when flies are loaded into *Drosophila* Activity Monitor (DAM) glass tubes (*Robles-Murguia et al., 2019*). We crossed the *nSyb-GS* flies with Polr1F or Regnase-1 RNAi flies and transferred the adult F1 progeny to RU486 tubes to induce expression of the RNAi 3–5 hr before dusk, and continuously monitored sleep for 5 days. Knockdown of Polr1F resulted in immediate inactivity and sleep (*Figure 3A–C*). However, with knockdown of Regnase-1, immediate changes in sleep were not noted (*Figure 3—figure supplement 2A-C*). This could be due to potential leakiness of the nSyb-GS, such that it reduced sleep even without RU486 treatment, coupled with the fact that sleep before dusk is already quite low, potentially hindering detection of further sleep reduction. We also analyzed sleep for the next 3 consecutive days from day 3 to day 5 and found that constitutive pan-neuronal Polr1F knockdown increased sleep while Regnase-1 knockdown seemed to have no effect, again perhaps due to the leakiness of *nSyb-GS*. However, it is also possible that Regnase-1 affects sleep during developmental stages or acts differently in different brain areas (*Figure 3D and E*; *Figure 3—figure supplement 2D, E*). In general, these results indicate that adult specific knockdown of Polr1F promotes sleep while brain-wide adult-specific knockdown of Regnase-1 has limited effect on sleep.

To further assess the adult-specific sleep function of both genes in ap α′/β′ neurons, we coupled the R35B12-Gal4 driver to tubulin-Gal80$^{ts}$, which allows temperature-dependent expression of Gal4, and crossed it with the UAS-Polr1F$^{RNAi}$ and UAS-Regnase-1$^{RNAi}$ lines. We confirmed that Polr1F RNAi promotes both transient and chronic sleep when flies were transferred from permissive to restrictive temperatures during the adult stages (*Figure 3—figure supplement 1*). Conversely, Regnase-1 RNAi showed no effect on sleep in the adult stage, consistent with our nSyb-GS experiments, suggesting that the Regnase-1 RNAi sleep effect is likely developmental (*Figure 3—figure supplement 3*). We also analyzed the daily peak activity of the flies to determine if either Polr1F RNAi or Regnase-1 RNAi affected activity. We found that neither gene affected peak activity, suggesting that both genes influence sleep but not activity (*Figure 3—figure supplement 4*).

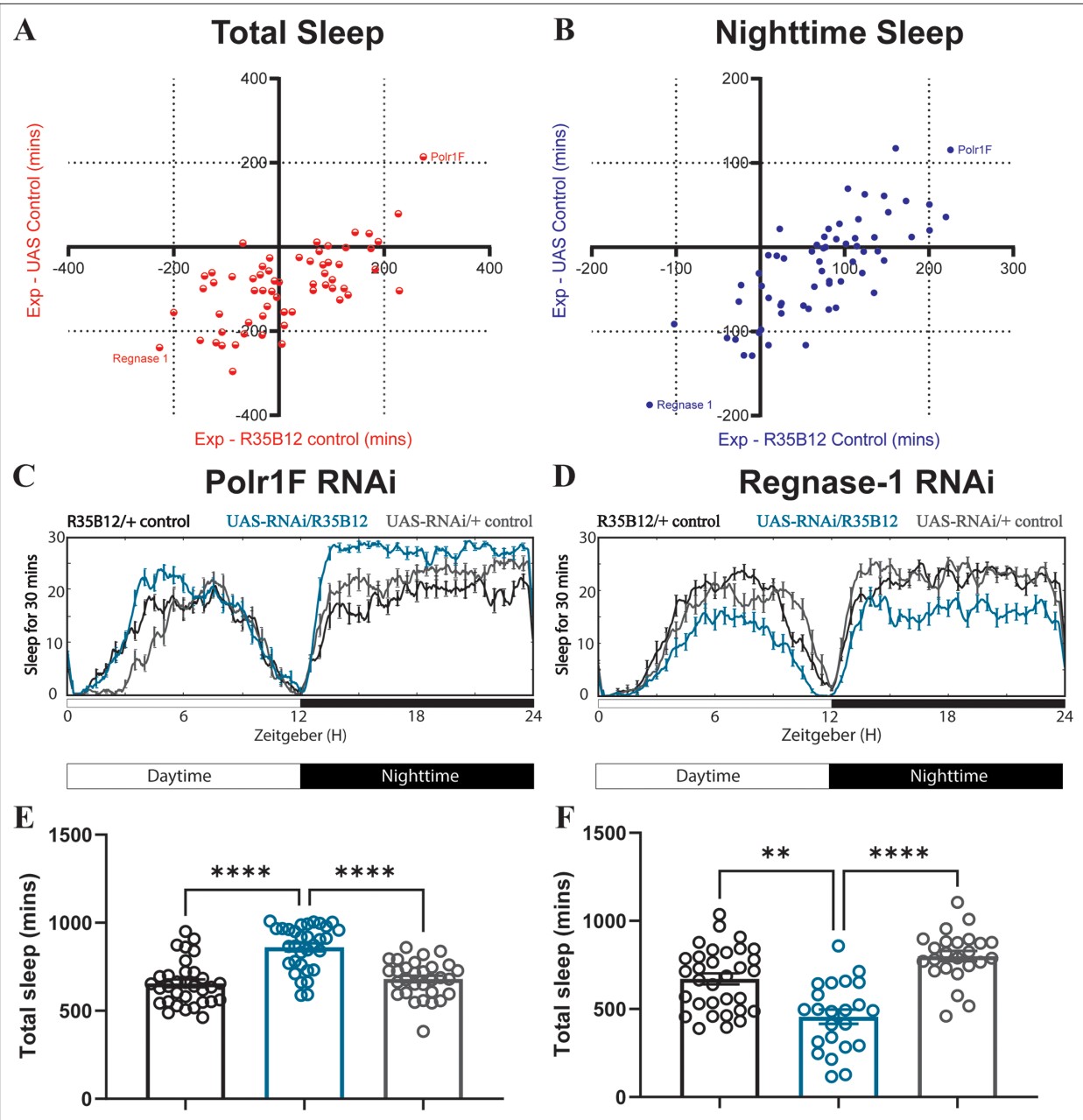

**Figure 2.** The sleep screen of differentially expressed genes identifies Polr1F and Regnase-1 as sleep-regulating genes. (**A–B**) Flies carrying the ap α'/β' neuron driver R35B12-Gal4 were crossed with flies carrying UAS-RNAi constructs targeting DEGs identified from RNA-seq analysis. 5–7 days old female F1 progeny were loaded onto Trikinetics DAM monitors to measure their sleep in a 12 hr light: 12 hr dark (12:12 LD) cycle. Mean total sleep (**A**) and nighttime sleep (**B**) were calculated by Pysolo and the difference between experimental flies and Gal4 and RNAi controls was calculated separately for each independent experiment; average values comparing each experimental to its Gal4 control (X-axis) and RNAi control (Y-axis) are shown in the plots. Of all the lines screened, knockdown of Polr1F and Regnase-1 had strongest effects on sleep, producing an increase and decrease in sleep, respectively. (**C–F**) show the representative sleep traces of *R35B12-Gal4>polr1F^RNAi* flies and *R35B12-Gal4>regnase1* RNAi. N=23–32 per genotype from two independent replicates combined are shown in E and F respectively, and bar graphs show mean + SEM. p values for each comparison were calculated using the Kruskal-Wallis test with Dunn's multiple comparisons test. **p<0.01, ***p<0.001, ****p<0.0001.

The online version of this article includes the following source data and figure supplement(s) for figure 2:

**Source data 1.** The sleep screen of differentially expressed genes and sleep phenotypes resulting from ap α'/β' neuron knockdown of Polr1F and Regnase-1.

**Figure supplement 1.** Effect of Polr1F and Regnase-1 knockdown on activity and sleep architecture.

**Figure supplement 1—source data 1.** Effect of Polr1F and Regnase-1 knockdown on activity and sleep architecture.

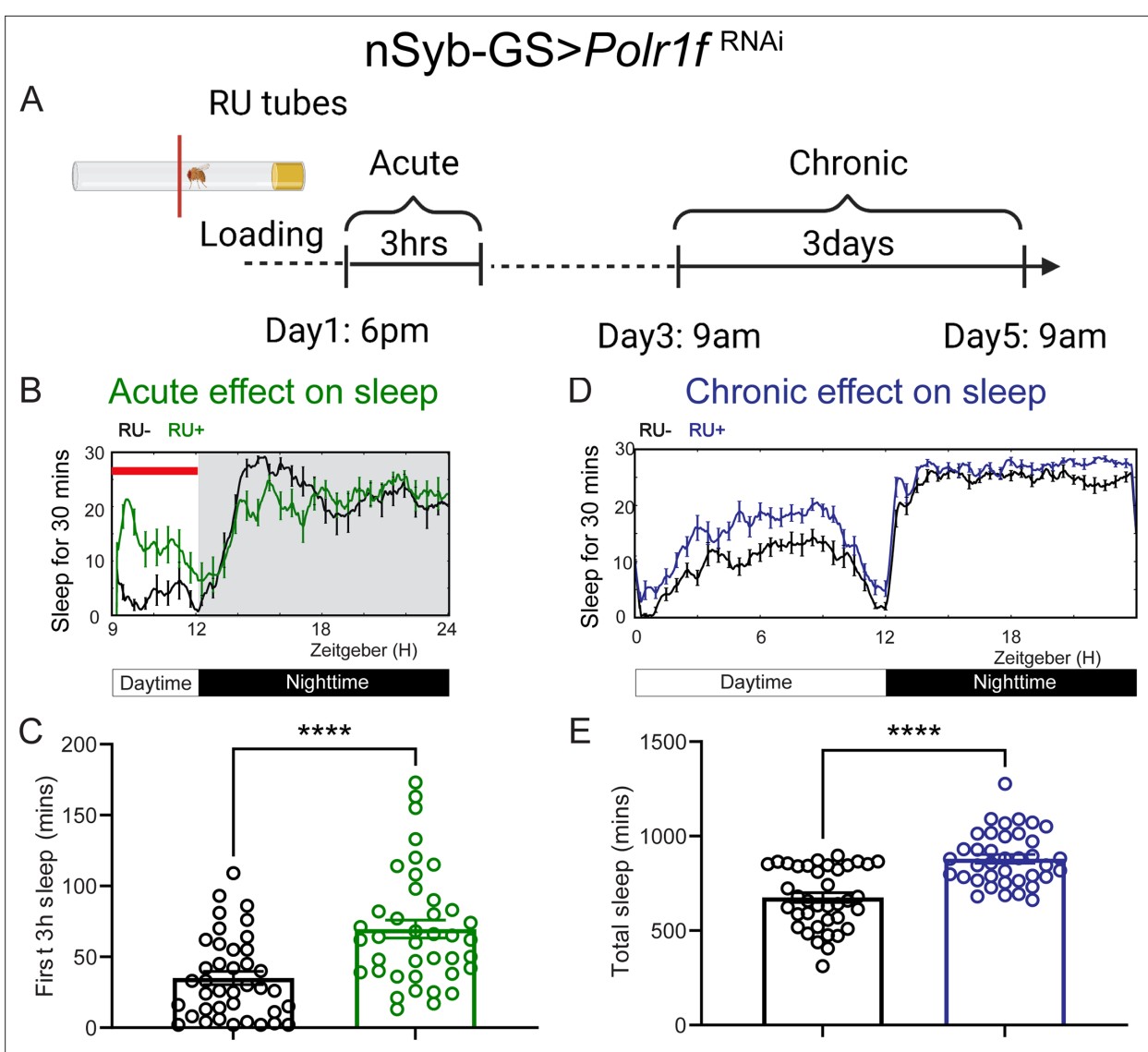

**Figure 3.** Acute and chronic effects of pan-neuronal knockdown of Polr1F on sleep in adult flies. (**A**) Schematic representation of transient and chronic sleep measurements in *nSyb-GS>polr1f^RNAi* flies (created in BioRender.com/t22a408). (**B**) Representative sleep traces and transient activity plot of flies expressing Polr1F RNAi under the control of an inducible pan-neuronal driver (*nSyb-GS>polr1f RNAi*) with and without RU treatment. (**C**) Quantification of sleep during the first 3 hours (ZT 9–12) after F1 progeny flies were loaded into RU- or RU+ DAM tubes at ZT8-T9. Sleep was measured starting at ZT9. N=39–40 individual flies per replicate with data from three independent replicates combined. The Mann-Whitney test was used to compare RU+ group and RU- groups. (**D**) Representative average sleep traces of *nSyb-GS>polr1f RNAi* in the RU- and RU+ DAM tubes for 3 consecutive days. Chronic sleep effects of pan-neuronal knockdown Polr1F were measured based on sleep data from day 3 to day 5. (**E**) Quantification of average total sleep of *nSyb-GS>polr1f^RNAi* and controls in the DAM tubes from (**D**). Unpaired t-test was used to compare between RU- and RU+ groups. ****p<0.0001.

The online version of this article includes the following source data and figure supplement(s) for figure 3:

**Source data 1.** Acute and chronic sleep phenotypes resulting from adult pan-neuronal knockdown of Porl1F.

**Figure supplement 1.** Adult specific knockdown of Polr1F promotes sleep.

**Figure supplement 1—source data 1.** Sleep phenotypes resulting from α'/β' neuronal knockdown of Polr1F by TARGET system.

**Figure supplement 2.** Acute and chronic effects of pan-neuronal knockdown of Regnase-1 on sleep in adult flies.

**Figure supplement 2—source data 1.** Acute and chronic sleep phenotypes resulting from adult pan-neuronal knockdown of Regnase-1.

**Figure supplement 3.** Adult specific knockdown of Regnase-1 has no effect on sleep.

**Figure supplement 3—source data 1.** Sleep phenotypes resulting from α'/β' neuronal knockdown of Regnase-1 by TARGET system.

**Figure supplement 4.** Adult specific knockdown of Polr1F or Regnase-1 has no effect on peak activity.

**Figure supplement 4—source data 1.** Effect of Polr1F and Regnase-1 knockdown by TARGET system on peak activity.

## Knockdown of Regnase-1 affects memory and post-training sleep

We next evaluated the impact of Polr1F and Regnase-1 knockdown on memory consolidation using our olfactory conditioning paradigm (*Figure 4A*). Starved flies were subjected to training to associate an odor with a reward, and then post-training, they were either kept on food vials for sleep-dependent memory consolidation or kept starved to promote sleep-independent memory consolidation. Memory tests were conducted 24 hr after training for starved flies, while fed flies were restarved for 42 hr before testing, as starvation is necessary for memory retrieval (*Krashes and Waddell, 2008*). Efficacy of the RNAi line used above was verified by qPCR experiments (*Figure 4—figure supplement 1A*). We observed that constitutive knockdown of Polr1F in ap α′/β′ neurons did not affect sleep-dependent or sleep-independent memory as memory performance was comparable to that of genetic controls (*Figure 4B–C*). These results were consistent with the fact that Polr1F levels typically decrease during memory consolidation. Monitoring of sleep from ZT8 to ZT12 after training at zeitgeber time (ZT) 6 showed that the post-training increase in sleep was also not affected by Polr1F knockdown in ap α′/β′ neurons (*Figure 4E*), suggesting that Polr1F does not have to be acutely downregulated for post-training sleep.

On the other hand, constitutive Regnase-1 knockdown in ap α′/β′ neurons resulted in a significant decrease in long-term memory performance in both fed and starved flies and eliminated the increase in post-training sleep (*Figure 4B–E*). These findings suggested that Regnase-1 expression in ap α′/β′ neurons is necessary for both sleep-dependent and sleep-independent memory consolidation. It is also possible that disruption of Regnase-1 in ap α′/β′ neurons affects learning, short-term memory formation, or appetitive memory retrieval (*Aso et al., 2014*; *Shyu et al., 2019*; *Rutherford et al., 2023*). Indeed, short-term memory tests confirmed that Regnase-1 knockdown flies performed significantly worse than control flies (*Figure 4F*). Given that the sleep effect of Regnase-1 is developmental, it is possible that this is also the case for the memory phenotype.

Regnase-1 overexpression had no effect on sleep (*Figure 4—figure supplement 1*; *Zhu et al., 2019*), but knockdown by R26E01-Gal4 in the m α′/β′ neurons also reduced sleep (data not shown), suggesting that Regnase-1 has a broader role that is not restricted to sleep-dependent memory.

## Knockdown of Polr1F promotes translation

Since Polr1F knockdown promotes sleep and a 22 amino acid peptide within Polr1F inhibits ribosomal DNA transcription (*Rothblum et al., 2014*), we predicted that Polr1F acted as a suppressor for ribosomal DNA transcription, and thus knockdown of Polr1F would enhance the transcription and translation of ribosomal RNA and thereby overall protein synthesis. As Regnase-1 is thought to promote decay of mRNAs undergoing translation, its knockdown, or its downregulation following training, might also be expected to promote translation, or at least the translation of its target mRNAs. The role of protein synthesis in long-term memory consolidation is well-established across organisms (*Alberini and Kandel, 2015*), and nascent rRNA synthesis was also shown to be induced by training and required for memory consolidation in mouse (*Allen et al., 2018*). We thus used real-time qPCR of total RNA extracted from vehicle control and RU-treated fly brains to measure how precursor ribosomal RNA (Pre-rRNA) is affected by knockdown of Polr1F. We found that the pre-rRNA level increased significantly in the RU486 induction (RU+) group compared with vehicle control (RU-) group for *nSyb-GS >polr1 F RNAi* flies (*Figure 5A*), indicating Polr1F knockdown results in higher levels of pre-rRNA, which is consistent with studies of Polr1F homologs in yeast cells (*Thuriaux et al., 1995*; *Rothblum et al., 2014*).

The increasing ribosomal RNA should help translation and so we also used incorporation of puromycin into newly synthesized peptides as an estimate of translation after inducing pan-neuronal knockdown of Polr1F. We observed significantly higher levels of anti-puromycin GFP fluorescence when Porl1F was knockdown pan-neuronally (RU+) compared to the control (RU-) group (*Figure 5B*), indicating that Polr1F suppresses translation, probably by suppressing ribosomal RNA transcription (*Rothblum et al., 2014*). Thus, it may need to be downregulated after training to support translation and memory. Given that knockdown of Porl1F enhances rRNA synthesis and sleep, there is a question as to whether global alterations in rRNA synthesis impact sleep. Feeding of an rRNA inhibitor (CX-5461) failed to produce rapid sleep phenotypes in fed or starved flies (*Figure 5—figure supplement 1*), but this negative result cannot be considered conclusive.

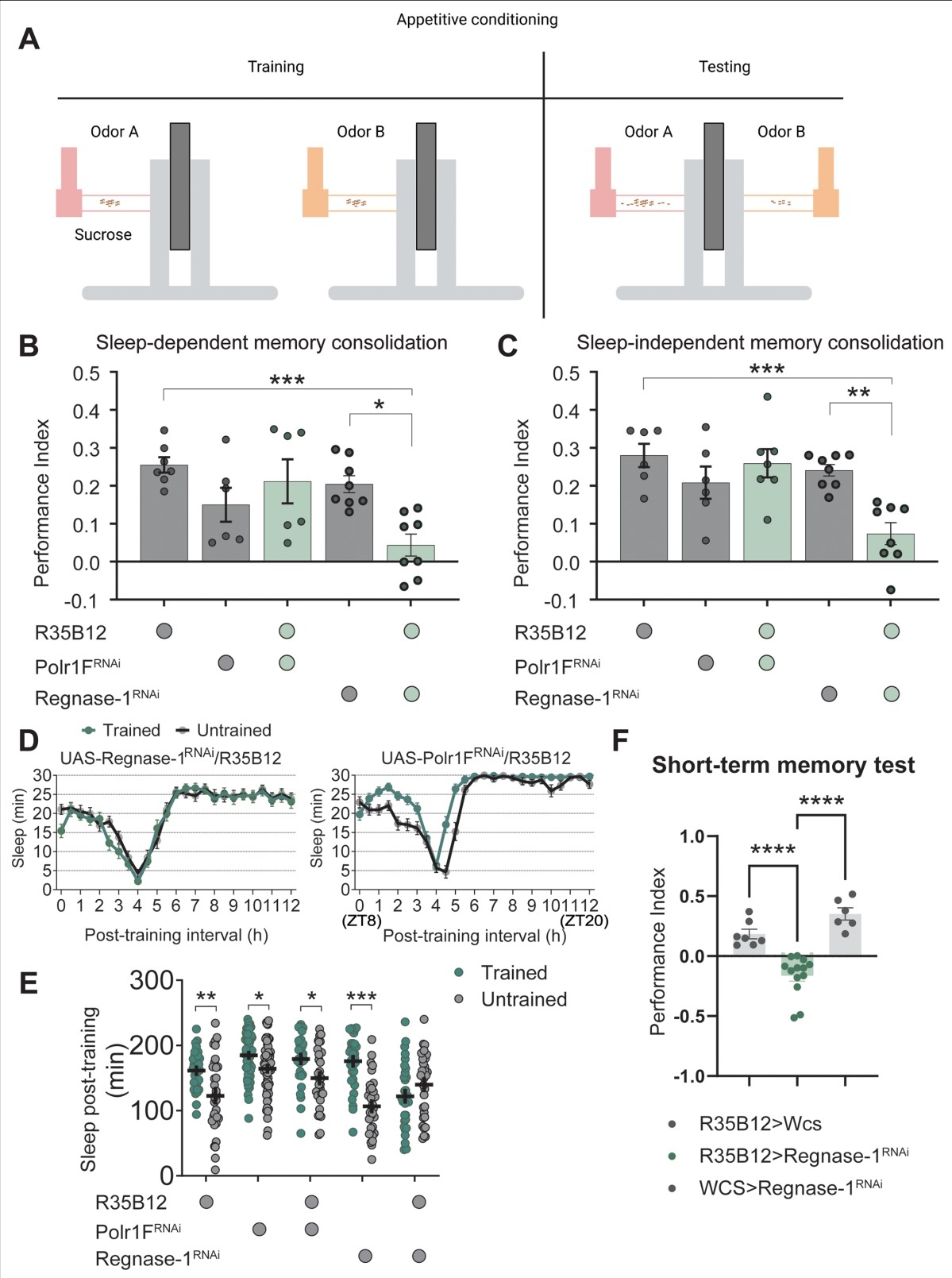

**Figure 4.** Regnase-1 expression is essential for sleep-dependent and sleep-independent memory. (**A**) Schematic representation of the memory test protocol (created in BioRender.com/z46i047). (**B, C**) Sleep-dependent and sleep-independent memory tests were conducted under fed and starved conditions, respectively. Knockdown of Regnase-1 significantly reduces long-term memory performance in both fed and starved flies. However, knockdown of Polr1F in ap α′/β′ neurons does not affect long-term memory performance. N≥6 biological replicates, each replicate containing 100–150

*Figure 4 continued on next page*

*Figure 4 continued*

flies. (**D, E**) Fed UAS-regnase-1-RNAi/+and R35B12/+flies exhibit a significant increase in sleep after training, while *R35B12-Gal4>regnase-1 RNAi* flies fail to show a comparable increase in post-training sleep. The total sleep in the ZT8-ZT12 interval is shown in (**E**). Polr1F knockdown in ap α′/β′ neurons does not affect the post-training increase in sleep. N≥32. (**F**) Compared to R35B12-Gal4/+and + /UAS-Regnase-1 RNAi flies, *R35B12-Gal4>reganse-1 RNAi* flies show a significant decrease in the performance index in short-term memory. N≥6 biological replicates, each containing 100–150 flies. ns = not significant, p>0.05, *p<0.05, **p<0.01, ***p<0.001, ****p<0.0001.

The online version of this article includes the following source data and figure supplement(s) for figure 4:

**Source data 1.** Effects of Porl1F and Regnase-1 knockdown on memory and post-training sleep.

**Figure supplement 1.** Regnase-1 overexpression in the ap α′/β′ neurons does not affect sleep.

**Figure supplement 1—source data 1.** Verification of RNAi line efficacy and the sleep phenotypes resulting from Regnase-1 overexpression.

Using puromycin to address effects of Regnase-1 knockdown on translation revealed an insignificant slight increase in translation (data not shown); given that Regnase-1 may specifically affect pre-existing translationally active mRNA or may act only on specific target mRNAs, its effects on de novo translation may not be obvious.

## Discussion

The anterior-posterior (ap) α′/β′ neurons of the mushroom body make critical and privileged contributions to sleep-dependent memory consolidation and post-training sleep (*Krashes et al., 2007*; *Chouhan et al., 2021*), but the mechanisms for sleep-dependent memory in these neurons are not known. To address this gap, we conducted transcriptomic analysis of ap α′/β′ neurons from trained and fed flies to identify genes that change rapidly under conditions that drive sleep-dependent memory. Our transcriptome profiling of ap α′/β′ neurons suggests that genes regulating rRNA transcription and translation are altered in the context of sleep-dependent memory consolidation.

### RNA processing genes mediate sleep-dependent memory consolidation and sleep

Many of the 59 DEGs we identified are implicated in RNA processing. Of these, Polr1F and CG11920 affect ribosomal RNA processing, CG5654 is predicted to be part of the 90 S pre-ribosome and involved in endonucleoytic cleavages (*Herold et al., 2009*), WDR79 encodes a small Cajal body specific RNA binding protein, Nup133 encodes a component of nuclear pore complex, Regnase-1 degrades mRNA, and CG18011, CG17568, Koko, CG11398 and Meics are all involved in RNA polymerase II-specific transcription (*Di Giorgio et al., 2017*; *Rogg et al., 2022*). The enrichment of RNA processing and translation genes is consistent with the notion that memory consolidation requires transcription and translation (*Seibt and Frank, 2012*; *Alberini and Kandel, 2015*). More importantly, our findings here indicate that alterations of some of RNA processing genes impact sleep as well.

### Polr1F regulates ribosome RNA synthesis and sleep

Learning-induced changes in gene expression in memory-related neurons are often critical for long-term memory formation (*Cavallaro et al., 2002*; *Hoedjes et al., 2015*; *Tadi et al., 2015*). Our findings with Polr1F implicate changes in Pol I transcription during sleep-dependent memory. Polr1F(Rpa43) is predicted to be part of the RNA polymerase I complex and is involved in DNA-dependent RNA polymerase activity/rDNA transcription in yeast, especially for the initiation of ribosomal RNA (*Beckouët et al., 2011*; *Marygold et al., 2020*). As suggested by our data, Polr1F has an inhibitory role in the RNA polymerase I complex, which is also consistent with the aforementioned study that found a 22 amino acid peptide within Polr1F can inhibit rDNA transcription (*Rothblum et al., 2014*). Based upon our finding that inducible knockdown of Polr1F rapidly promotes translation, it is likely that the rapid and dramatic decline of Polr1F after training in fed flies serves to increase de novo ribosome RNA synthesis. This supports the report that ribosomal RNA is induced by learning and required for memory consolidation in mice (*Allen et al., 2018*). While we do not know how knockdown of Polr1F promotes sleep, an attractive possibility is that higher translation is a result of elevated sleep. Sleep is thought to promote translation and its role in memory consolidation in fed flies could be related to its effect on translation (*Seibt and Frank, 2012*; *Chouhan et al., 2021*). Alternatively, increased

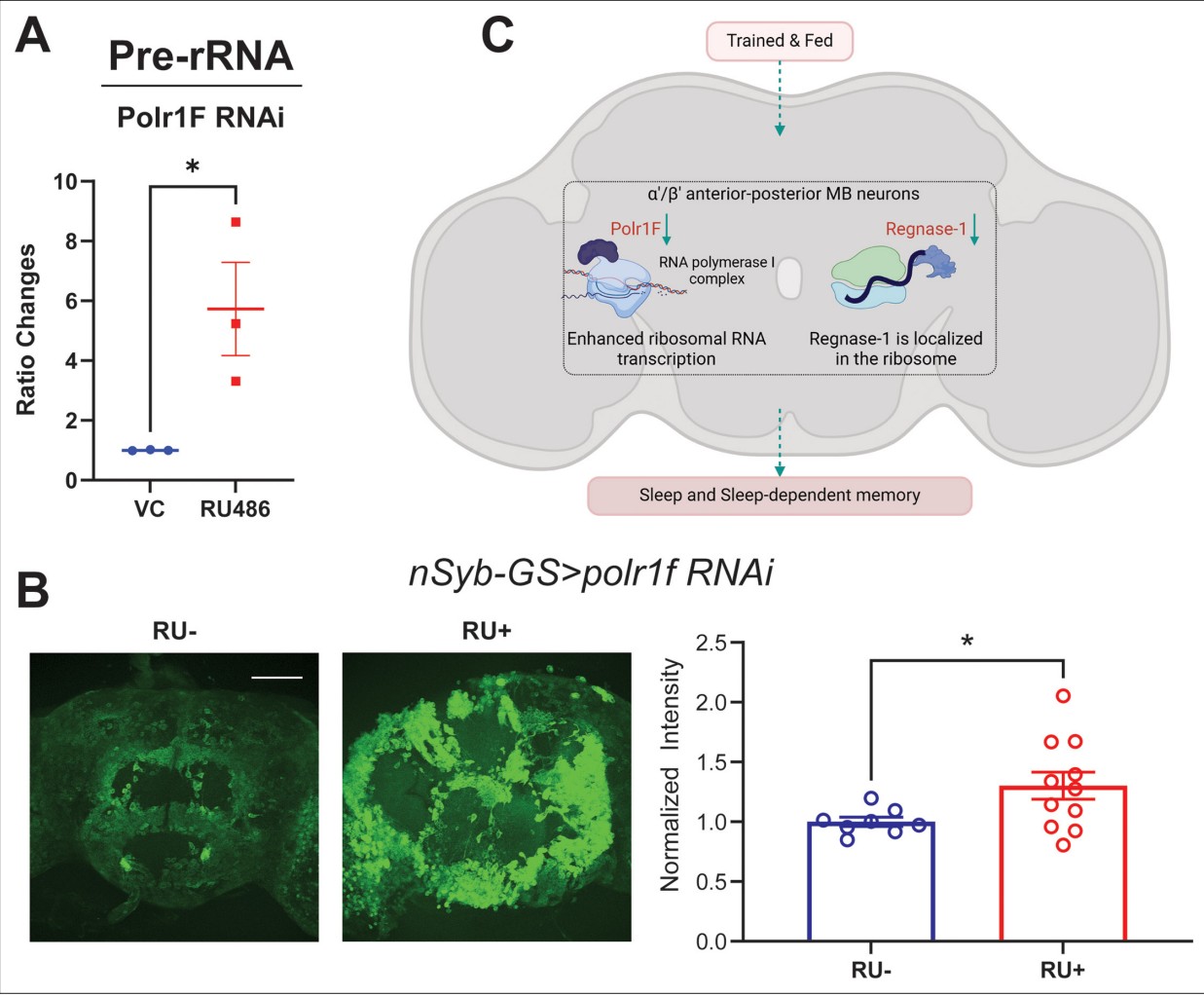

**Figure 5.** Knockdown of Polr1F results in high translation. (**A**) The *nSyb-GS >polr1 f RNAi* flies exhibit a significant increase in pre-rRNA levels. (**B**) The ex vivo puromycin immunostaining assay was used to measure translation in dissected whole brains. The results show that knockdown of Polr1F using the pan-neuronal nSyb-GeneSwitch (GS) system increases translation relative to control flies that were not treated with RU. The normalized mean grayscales from the RU- and RU +groups are compared using an unpaired t-test. The analysis includes data from 8 to 11 flies per group, with results from two independent replicates combined. Scale bar: 100 µm. (**C**) The schematic model (created in BioRender.com/h27z926) illustrates the major transcriptome features of ap α'/β' neurons under trained and fed conditions, revealed in this manuscript. Two RNA processing genes Polr1F and Regnase-1 are prominently downregulated during memory consolidation in trained and fed flies, respectively. Polr1F is involved in regulating ribosomal RNA synthesis, and its decrease in levels in trained and fed flies promotes sleep and translation. In contrast, Regnase-1 is localized in the ribosome and involved in mRNA decay, and its downregulation causes deficits in sleep and memory.

The online version of this article includes the following source data and figure supplement(s) for figure 5:

**Source data 1.** The quantification of pre-rRNA levels and puromycin GFP signals resulting from Polr1F knockdown.

**Figure supplement 1.** rRNA inhibitor (CX-5461) feeding does not affect sleep.

**Figure supplement 1—source data 1.** Sleep phenotypes resulting from rRNA inhibitor (CX-5461) feeding.

translation or rRNA synthesis could promote sleep. However, translation is typically thought of as a consequence of sleep rather than a cause (*Zimmerman et al., 2006*). Also, rRNA transcription rates remain constant throughout the day in the liver (*Sinturel et al., 2017*), but it is still possible that these rates vary in particular regions of the brain and affect sleep. The role of rRNA synthesis in *Drosophila* learning and memory has barely been explored, but our work, together with that of *Allen et al., 2018*, indicates that the well-known requirement for de novo protein synthesis during long-term memory consolidation (*Jarome and Helmstetter, 2014*) includes increased synthesis of ribosomal RNA and protein.

## Novel neuronal role of Regnase-1 in sleep and memory

The role of RNA binding protein Regnase-1 in the innate immune response has been extensively studied (*Mino et al., 2015*; *Mao et al., 2017*; *Wei et al., 2019*). However, our study sheds light on a novel neuronal function of Regase-1 on sleep and memory. Regnase-1 is an anti-inflammatory enzyme that inhibits mRNA translation during acute inflammatory responses. It localizes to the ribosomes on the surface of the endoplasmic reticulum (ER) and binds to translationally active mRNAs with specialized stem-loop structures at the 3'UTR (*Uehata et al., 2013*; *Mino et al., 2015*). When phosphorylated, Regnase-1 is released from the ER (*Tanaka et al., 2019*). Because functional Regnase-1 binds and degrades its bound mRNA, Regnase-1 inactivation leads to an increase of its target mRNA (*Uehata et al., 2013*). The target mRNAs of Regnase-1 in immune cells encode proinflammatory cytokines, which can then be expressed when Regnase-1 is inactivated. However, Regnase-1 has also been reported to modulate cytokines and neuronal injury in the microglia in rats (*Liu et al., 2016*). Regulation of Regnase-1 is usually rapid and transient, and its rapid response to microenvironmental changes, different pathological states and stress is critical for cellular adaption (*Mao et al., 2017*).

Our study reveals that the expression of Regnase-1 changes after training, and constitutive downregulation of Regnase-1 in ap α'/β' neurons reduces sleep and causes deficits in learning and memory consolidation. Whether acute downregulation of Regnase-1 is necessary for sleep-dependent memory remains unclear, but it could play a role by promoting the translation of specific transcripts. For instance, post-training downregulation of Regnase-1 could release mRNAs that are usually targeted for decay but are critical for memory consolidation. Constitutive loss of Regnase-1 impairs sleep-independent memory and short-term memory, although the mechanisms are not known. We note that ap α'/β' neurons have a role in several aspects of memory consolidation as well as short-term appetitive memory retrieval. Interestingly, constitutive knockdown of Regnase-1 also reduces sleep developmentally and prevents sleep increase after training in adult. While these could be independent effects, it is also possible that loss of Regnase-1 affects the development of the relevant MB neurons, thereby impacting all behaviors regulated by these neurons.

Our study of local molecular changes in ap α'/β' neurons after training suggests that ribosomal RNA transcription and mRNA translation might work in concert during the consolidation of sleep-dependent memory. How sleep is involved in this boosted protein synthesis process is unclear, but we suggest that RNA processing changes induce sleep, which promotes translation necessary for consolidation of long-term memory. However, these processes need to be teased apart experimentally. Overall, our findings demonstrate a role of RNA processing in sleep and memory, providing a foundation for future exploration of the mechanisms involved.

## Materials and methods

**Key resources table**

| Reagent type (species) or resource | Designation | Source or reference | Identifiers | Additional information |
|---|---|---|---|---|
| Genetic reagent (*D. melanogaster*) | white Canton-S (wCS) | Laboratory Stocks | | |
| Genetic reagent (*D. melanogaster*) | nSyb-GS | Laboratory Stocks | PMID:29590612 | |
| Genetic reagent (*D. melanogaster*) | R35B12-Gal4 | Bloomington Stock Center | 49822 | |
| Genetic reagent (*D. melanogaster*) | UAS-nGFP | Bloomington Stock Center | 4775 | |
| Genetic reagent (*D. melanogaster*) | R26E01-Gal4 | Bloomington Stock Center | 60510 | |
| Genetic reagent (*D. melanogaster*) | UAS-Polr1F RNAi | Bloomington Stock Center | 64553 | |
| Genetic reagent (*D. melanogaster*) | UAS-Regnase-1 RNAi | VDRC Stock Center | 27330 | |

*Continued on next page*

*Continued*

| Reagent type (species) or resource | Designation | Source or reference | Identifiers | Additional information |
|---|---|---|---|---|
| Genetic reagent (*D. melanogaster*) | UAS-Rengase-1 OE | Ryuya Fukunaga lab | | |
| Genetic reagent (*D. melanogaster*) | UAS-Polr1F RNAi (Used in **Figure 3—figure supplements 1 and 4**) | VDRC Stock Center | 103392 | |
| Antibody | Anti-Puromycin [3RH11] Antibody, mouse monoclonal | Kerafast | EQ0001 | IF(1:1000) |
| Sequence-based reagent | Pre-rRNA oligo | F: ATG GCC GTA TTC GAA TGG ATT TA | This paper | |
| Sequence-based reagent | Pre-rRNA oligo | R: CTA CTG GCA GGA TCA ACC AGA | This paper | |
| Sequence-based reagent | Regnase-1 oligo | F: CAG TCC GGG TGG CAA TAA TA | This paper | |
| Sequence-based reagent | Regnase-1 oligo | R: AGA TCC ATT TGA GCG GAG AAG | This paper | |
| Sequence-based reagent | Polr1F oligos | F: GGG TCT TCA ACA CCT CCA TAC | This paper | |
| Sequence-based reagent | Polr1F oligos | R: GCA ATA GTT CTC CCT CGA TGT AA | This paper | |
| Sequence-based reagent | Rp49 oligo | F: CCA CCA GTC GGA TCG ATA TGC | This paper | |
| Sequence-based reagent | Rp49 oligo | R: CTC TTG AGA ACG CAG GCG ACC | This paper | |
| Software, algorithm | GraphPad Prism v9 | GraphPad Software | RRID:SCR_002798 | |
| Software, algorithm | DAMFileScan113 | Trikinetics | https://trikinetics.com/ | |
| Software, algorithm | Pysolo | *Gilestro and Cirelli, 2009* | https://www.pysolo.net/about/ | |
| Software, algorithm | Adobe Illustrator 2020 | Adobe | https://www.adobe.com/ | |
| Software, algorithm | BioRender | BioRender | RRID:SCR_018361 | |
| Software, algorithm | Fiji software | ImagJ | RRID:SCR_002285 | |
| Chemical compound, drug | Schneider's medium | Thermofisher | 21720024 | |
| Chemical compound, drug | Papain | Worthington PAP2 | LK003178 | |
| Chemical compound, drug | Liberase | Roche | 5401119001 | 2.5 mg/ml |
| Chemical compound, drug | DAPI | Thermofisher | 62247 | |
| Chemical compound, drug | Puromycin dihydrochloride | Santa Cruz | Sc-108071A | |
| Chemical compound, drug | RNA Polymerase I Inhibitor II, CX-5461 | Sigma | 5092650001 | 0.2 mM |
| Commercial assay or kit | RNeasy Plus Mini Kit | Qiagen | Item No. 74134 | |

## Contact for reagent and resource sharing

Amita Sehgal (amita@pennmedicine.upenn.edu).

## Fly stock and maintenance

All the stock information of the flies used in this project are listed in the key resource table and flies were reared on the standard cornmeal vials or bottles at 25 °C with 12:12 hr light dark cycle in the preset incubator. The genetic background control used in the paper is White-CantonS (wCS) unless specified.

## Behavior measurement in *Drosophila*

We have used both single beam and updated multibeam *Drosophila* activity monitoring (DAM) system from Trikinetics (https://trikinetics.com/) in our experiments. Briefly, 5–7 days old female flies were loaded into 60/90 mm glass locomotor tubes for behavior tests, using DAM2/5 H *Drosophila* activity monitors from Trikinetics. 1/15 infrared beams bisect each tube, providing movement (position in multibeam) information of the fly across the tube. Locomotor tubes are loaded with 2% agar with 5% sucrose as fly food on one side, and yarn is put on the other side to restrain the behavior of flies inside the glass tubes. For experiments with the inducible Gene Switch system, 0.5 mM RU486 (mifepristone) was added to the fly food to activate the expression of the transgenes under the control of UAS. Three constitutive days of data were used for sleep analysis by Pysolo (https://www.pysolo.net/).

## Appetitive conditioning

Approximately 100 four- to seven-day-old mixed-sexes flies were starved for 12 hr in *Drosophila* bottles with water -soaked filtered paper and then trained at 25 °C and 70% relative humidity to associate sucrose with odor A for 2 min, and then a blank with odor B for 2 min with 30 s clean air in between. The odors used were 4-Methylcyclohexanol (MCH) and 3-Octanol (OCT) in paraffin oil and the concentration for both MCH and OCT in oil was 1:10. After conditioning, flies were moved back to normal fly food or starved for 1 hr and dissected for subsequent ap cell sorting and RNA-sequencing. For the short-term memory test, flies were tested immediately after conditioning in the same wheel for 2 min.

To assess post-training sleep, flies were introduced in glass tubes containing 2% agar and 5% sucrose through an aspirator without anesthesia and loaded into the DAM system after training. For long-term memory assessment, trained flies were either kept on food vials for 24 hr or were further starved. Starved flies were tested for memory 24 hr after training, while fed flies were re-starved for 42 hr before memory tests. Memory was tested by giving flies a choice between odor A and odor B for 2 min in a T-maze. Performance index (PI) was calculated as the number of flies selecting CS$^+$ odor minus the number of flies selecting CS$^-$ odor divided by the total number of flies. Each PI is the average of PIs from reciprocal experiments with two odors swapped to minimize non-associative effects.

## Cell isolation and sorting

Dissected brains are dissociated by following the protocol from *Li et al., 2017*. Briefly, brains are dissected in Schneider's medium, and then are placed in a shaker and dissociated in Papain solution, filtered through a 100 μm cell strainer, and re-suspended in Schneider's medium. 500 GFP+ cells from the same conditions were sorted into 96 well microplate with lysis buffer from Smart-seq2 HT kit and frozen. We dissected 50 brains for each group to ensure enough GFP+ cells. Cell sorting were conducted by either BD FACSMelody or BD FACSAria (BD Biosciences), and dead cells were excluded with 4′, 6-diamidino-2-phenylindole (DAPI). Doublets were excluded using and forward scatter (FSC-H by FSC-W) and side scatter (SSC-H by SSC-W). Size of cells was selected by FSC-A by FSC-A and validated for fly neurons using cells from flies expressing nsyb-nGFP. Length of time from tissue harvest to cell collection approximated 4hours.

## RNA-seq and data analysis

GFP+ cells were sorted and immediately frozen, then sent to Admera Health (https://www.admera-health.com/) for RNA extraction, RNA library construction, and sequencing using the Smart-seq2 HT kit. To analyze the RNA-sequence data, we used Hisat2 v2.2.1, RRID:SCR_015530 (http://daehwank-imlab.github.io/hisat2/) to map the sequencing data FASTQ files to the fly genome (BDSG6) (*Kim et al., 2019*). The alignment results were then counted by LiBiNorm (https://warwick.ac.uk/fac/sci/lifesci/research/libinorm/) using the GENCODE reference genome (*Dyer et al., 2019*). Raw count and TPM were used separately in further analysis. Raw count data were analyzed by IDEP v0.95 to identify genes expressed differentially between three conditions (*Ge et al., 2018*). We filtered out low-expressed genes using a cutoff of CPM >0.5, at least detected in three independent samples, and treated missing values as gene median. Regularized log transformation was used to transform raw count data for clustering and PCA. Differentially expressed genes were identified using DESeq2 with an FDR cutoff of 0.1 and minimum fold change of 2.

## Puromycin assay and imaging

We developed a puromycin assay to measure the rate and localization of nascent peptide synthesis in the fly brain, and similar method has been described in the fly larvae (*Deliu et al., 2017*). Fly brains were dissected in Schneiders' medium and incubated with puromycin for 40 min in vitro to allow puromycin to incorporate into the newly synthesized peptide. Subsequently, using an anti-puromycin antibody, standard immunostaining protocols were applied to detect the number and position of newly synthesized puromycin-tagged peptides (*Aviner, 2020*), which provided an estimate of translation rate. The brains were then imaged with a 20 X oil immersion confocal microscope with a resolution of 1024*1024. The GFP intensity of the images was then measured by Fiji software (ImageJ) and the average intensity of the samples was analyzed for comparison.

## RNA polymerase I inhibitor II, CX-5461 protocol

CX-5461 was mixed with 2% agar and 5% sucrose to make fly motor tubes at a final concentration of 0.2 mM. Flies are loaded into these CX-5461 tubes or vehicle control tubes 4 hr before light turns off and transient sleep changes are measured.

## Quantitative real-time PCR (qPCR)

10 flies' brains from each group were dissected for RNA extraction using QIAGEN's RNeasy Plus Mini Kit. Total RNA was then reverse transcribed to cDNA by using random hexamers and Superscript II from Invitrogen. qPCR was performed using SYBR-green master mix and oligonucleotide information is provided in the Key Resources table. Relative gene expression was analyzed using the ΔΔCt method.

## Statistical analysis

Fly sleep behavioral data extracted from Pysolo was analyzed by GraphPad Prism (https://www.graphpad.com/). Data from different replicates were pooled directly and first tested for normality using D'Agostino-Pearson and Shapiro-Wilk tests. For normally distributed data, unpaired parametric Student's t-test is used for two-sample experiments and one-way ANOVA with Turkey post hoc test for three-sample or more experiments. Non-normally distributed data were analyzed using nonparametric tests, the Mann-Whitney test for two-sample experiments and the Kruskal-Wallis test with Dunn's multiple comparisons test for three or more samples experiments. For all graphs, unless otherwise stated, data are presented as mean and standard error of the mean (SEM) and statistical significance was accepted for -value <0.05.

# Acknowledgements

This work was supported by Howard Hughes Medical Institute (HHMI). We thank all members of the Sehgal lab for reagents, comments and support, especially rotation undergraduate Arielle Ketchum for help with molecular cloning. We thank Hongjie Li of Liqun Luo lab with technical assistance and protocol sharing from Stanford University. We thank Ryuya Fukunaga lab for kindly sharing with us the Regnase-1 overexpression line.

# Additional information

### Funding

| Funder | Grant reference number | Author |
|---|---|---|
| Howard Hughes Medical Institute | | Amita Sehgal |

The funders had no role in study design, data collection and interpretation, or the decision to submit the work for publication.

### Author contributions

Yongjun Li, Conceptualization, Data curation, Software, Formal analysis, Validation, Investigation, Visualization, Methodology, Writing – original draft, Writing – review and editing; Nitin S Chouhan,

Conceptualization, Data curation, Formal analysis, Validation, Investigation, Visualization, Methodology, Writing – review and editing; Shirley L Zhang, Conceptualization, Software, Validation, Methodology; Rebecca S Moore, Resources, Data curation, Formal analysis, Validation, Visualization, Methodology; Sara B Noya, Data curation, Validation, Methodology; Joy Shon, Resources, Software, Validation, Visualization, Methodology; Zhifeng Yue, Resources, Validation, Methodology; Amita Sehgal, Conceptualization, Resources, Formal analysis, Supervision, Funding acquisition, Investigation, Project administration, Writing – review and editing

### Author ORCIDs
Yongjun Li ⬤ http://orcid.org/0000-0002-3618-6041
Shirley L Zhang ⬤ https://orcid.org/0000-0002-6672-2044
Rebecca S Moore ⬤ https://orcid.org/0000-0003-3155-3436
Amita Sehgal ⬤ https://orcid.org/0000-0001-7354-9641

Reviewer #2 (Public review): https://doi.org/10.7554/eLife.89023.4.sa1
Reviewer #3 (Public review): https://doi.org/10.7554/eLife.89023.4.sa2
Reviewer #4 (Public review): https://doi.org/10.7554/eLife.89023.4.sa3
Author response https://doi.org/10.7554/eLife.89023.4.sa4

## Additional files

### Supplementary files
• MDAR checklist

### Data availability
Sequencing data have been deposited in GEO under accession code PRJNA1132369. All other data generated or analyzed during this study are included in the manuscript and supporting files.

The following dataset was generated:

| Author(s) | Year | Dataset title | Dataset URL | Database and Identifier |
|-----------|------|---------------|-------------|-------------------------|
| Sehgal A | 2024 | Transcriptomic analysis of mushroom anterior-posterior neurons for memory and sleep | http://www.ncbi.nlm.nih.gov/bioproject/?term=PRJNA1132369 | NCBI BioProject, PRJNA1132369 |

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
