## [Editor Report · eLife Assessment]

The aim of this **important** study is to identify novel genes involved in sleep regulation and memory consolidation. It combines transcriptomic approaches following memory induction with measurements of sleep and memory to discover molecular pathways underlying these interlinked behaviors. The authors explore transcriptional changes in specific mushroom body neurons and suggest roles for two genes involved in RNA processing, Polr1F and Regnase-1, in the regulation of sleep and memory. Their findings offer **convincing** evidence that the expression of RNA processing genes is modulated during sleep-dependent memory, with Polr1F potentially contributing to increased sleep.

---

## [Referee Report · Reviewer #2 (Public review)]

Sleep and memory are intertwined processes, with sleep-deprivation having a negative impact on long-term memory in many species. Recently, the authors showed that fruit flies form sleep-dependent long-term appetitive memory only when fed. They showed that this context-dependent memory trace maps to the anterior posterior (ap) α'β' mushroom body neurons (MBNs) (Chouhan et al., (2021) Nature). However, the molecular cascades induced by during training that promote sleep and memory have remained enigmatic.

Here the authors investigate this issue by combining cell-specific transcriptomics, genetic perturbations, and measurements of sleep and memory. They identify an array of genes altered in expression following appetitive training. These genes are mainly downregulated, and predominantly encode regulators of transcription and RNA biosynthesis. This is a conceptually attractive finding given that long-term memory requires de novo protein translation.

The authors then screen these genes for novel regulators of sleep and memory. They show that one of these genes (Polr1F) acts in ap α'β' MBNs to promote wakefulness, while another (Regnase-1) promotes sleep. They also identify a specific role for Regnase-1 in ap α'β' MBNs in regulating short- and long-term memory formation - likely through effects on the development of ap α'β' MBNs - and demonstrate that Pol1rF inhibits translation throughout the fly brain.

The analyses of molecular alterations in ap α'β' MBNs are interesting and impressive. However, as noted by the authors, further experiments are required to clarify the precise contribution of reductions in Polr1F and Regnase-1 to training-induced changes in memory and sleep. Nonetheless, this study provides a useful platform for such studies, and provides a conceptual advance in linking acute changes in RNA processing pathways to the interconnected processes of sleep, memory, and protein translation.

---

## [Referee Report · Reviewer #3 (Public review)]

Previous work (Chouhan et al., 2022) from the Sehgal group investigated the relationship between sleep and long-term memory formation by dissecting the role of mushroom body intrinsic neurons, extrinsic neurons, and output neurons during sleep-dependent and sleep-independent memory consolidation. In this manuscript, Li et al., profiled transcriptome in the anterior-posterior (ap) α'/β' neurons and identified genes that are differentially expressed after training in fed condition, which supports sleep-dependent memory formation. By knocking down candidate genes systematically, the authors identified Polr1F and Regnase-1 as two important hits that play potential roles in sleep and memory formation. What is the function of sleep and how to create a memory are two long-standing questions in science. The present study used a new approach to identify novel components that may link sleep and memory consolidation in a specific type of neuron. Importantly, these components implicated that RNA processing may play a role in these processes.

I am enthusiastic about the innovative approach employed to identify RNA processing genes involved in sleep regulation and memory consolidation. During the revision process, the authors fully addressed major concerns raised by reviewers. First, the author used the Gal80ts to restrict the knockdown of Regnase-1 in adult animals and concluded that Regnase-1 RNAi appears to affect sleep through development. Second, the author showed that Regnase-1 knockdown produced robust phenotypes for both sleep-dependent and sleep-independent memory, as well as a severe short-term memory phenotype. The author cautiously concluded that flies with constitutive Regnase-1 knockdown could be poor learners, thereby exhibiting a memory phenotype. Although we don't yet have a strong link between sleep and long-term memory consolidation, the interpretation presented in the manuscript is sufficiently justified by the data. This work presents a novel strategy to explore the link between sleep and memory consolidation.

---

## [Referee Report · Reviewer #4 (Public review)]

Summary:

Li and Chouhan et al. follow up on a previous publication describing the role of anterior-posterior (ap) and medial (m) ɑ′/β′ Kenyon cells in mediating sleep-dependent and sleep-independent memory consolidation, respectively, based on feeding state in *Drosophila melanogaster*. The authors sequenced bulk RNA of ap ɑ′/β′ Kenyon cells 1h after flies were either trained-fed, trained-starved or untrained-fed and find a small number of genes (59) differentially expressed (3 upregulated, 56 downregulated) between trained-fed and trained-starved conditions. Many of these genes encode proteins involved in the regulation of gene expression. The authors then screened these differentially expressed genes for sleep phenotypes by expressing RNAi hairpins constitutively in ap ɑ′/β′ Kenyon cells and measuring sleep patterns. Two hits were selected for further analysis: Polr1F, which promoted sleep, and Regnase-1, which reduced sleep. The pan-neuronal expression of Polr1F and Regnase-1 RNAi constructs was then temporally restricted to adult flies using the GeneSwitch system. Polr1F sleep phenotypes were still observed, while Regnase-1 sleep phenotypes were not, indicating developmental defects. Appetitive memory was then assessed in flies with constitutive knockdown of Polr1F and Regnase-1 in ap ɑ′/β′ Kenyon cells. Polr1F knockdown did not affect sleep-dependent or sleep-independent memory, while Regnase-1 knockdown disrupted sleep-dependent memory, sleep-independent memory, as well as learning. Polr1F knockdown increased pre-ribosomal RNA transcripts in the brain, as measured by qPCR, in line with its predicted role as part of the RNA polymerase I complex. A puromycin incorporation assay to fluorescently label newly synthesized proteins also indicated higher levels of bulk translation upon Polr1F knockdown. Regnase-1 knockdown did not lead to observable changes in measurements of bulk translation.

Strengths:

The proposed involvement of RNA processing genes in regulating sleep and memory processes is interesting, and relatively unexplored. The methods are satisfactory.

Weaknesses:

The main weakness of previous versions of the paper was the over-interpretation of results, particularly relating to the proposed link between sleep and memory consolidation. This has now been appropriately addressed, as reflected in the change of title and incorporation of alternative interpretations of the data in the text.

---

## [Author Response]

The following is the authors’ response to the previous reviews.

**eLife Assessment**
The aim of this valuable study is to identify novel genes involved in sleep regulation and memory consolidation. It combines transcriptomic approaches following memory induction with measurements of sleep and memory to discover molecular pathways underlying these interlinked behaviors. The authors explore transcriptional changes in specific mushroom body neurons and suggest roles for two genes involved in RNA processing, Polr1F and Regnase-1, in the regulation of sleep and memory. Although this work exploits convincing and validated methodology, the strength of the evidence is incomplete to support the main claim that these two genes establish a definitive link between sleep and memory consolidation.

We appreciate the reconsideration of our manuscript and recognize that we should have toned down the claims, especially with respect to the link between sleep and memory consolidation. We have now changed the title, the abstract and the main text and also Figure 5 to essentially just state our findings. While there is a little speculation in the Discussion, we point out that future work would be required to draw conclusions. We believe the manuscript still represents a considerable advance in showing the modulation of RNA processing genes during sleep-dependent memory consolidation in the relevant neurons, and also showing how one such gene affects sleep and translation and a second affects sleep and memory.

**Public Reviews:**

**Reviewer #2 (Public review):**
Prior work by the Sehgal group has shown that a small group of neurons in the fly brain (anterior posterior (ap) α'β' mushroom body neurons (MBNs)) promote sleep and sleep-dependent appetitive memory specifically under fed conditions (Chouhan et al., (2021) Nature). Here, Li, Chouhan et al. combine cell-specific transcriptomics with measurements of sleep and memory to identify molecular processes underlying this phenomenon. They define transcriptional changes in ap α'β' MBNs and suggest a role for two genes downregulated following memory induction (Polr1F and Regnase-1) in regulating sleep and memory.The transcriptional analyses in this manuscript are impressive. The authors have now included additional experiments that define acute and developmental roles for Polr1F and Regnase-1 respectively in regulating sleep. They have also provided additional data to strengthen their conclusion that Polr1F knockdown in α'β' mushroom body neurons enhances sleep.The resubmitted work represents a convincing investigation of two novel sleep-regulatory proteins that may also play important roles in memory formation.The authors have comprehensively addressed my comments, which I very much appreciate. I congratulate them on this excellent work.

We very much appreciate the reviewer’s positive feedback. Thank you!

**Reviewer #3 (Public review):**
Previous work (Chouhan et al., 2022) from the Sehgal group investigated the relationship between sleep and long-term memory formation by dissecting the role of mushroom body intrinsic neurons, extrinsic neurons, and output neurons during sleep-dependent and sleep-independent memory consolidation. In this manuscript, Li et al., profiled transcriptome in the anterior-posterior (ap) α'/β' neurons and identified genes that are differentially expressed after training in fed condition, which supports sleep-dependent memory formation. By knocking down candidate genes systematically, the authors identified Polr1F and Regnase-1 as two important hits that play potential roles in sleep and memory formation. What is the function of sleep and how to create a memory are two long-standing questions in science. The present study used a new approach to identify novel components that may link sleep and memory consolidation in a specific type of neuron. Importantly, these components implicated that RNA processing may play a role in these processes.While I am enthusiastic about the innovative approach employed to identify RNA processing genes involved in sleep regulation and memory consolidation, I feel that the data presented in the manuscript is insufficient to support the claim that these two genes establish a definitive link between sleep and memory consolidation. First, the developmental role of Regnase-1 in reducing sleep remains unclear because knocking down Regnase-1 using the GeneSwitch system produced neither acute nor chronic sleep loss phenotype. In the revised manuscript, the author used the Gal80ts to restrict the knockdown of Regnase-1 in adult animals and concluded that Regnase-1 RNAi appears to affect sleep through development. Conducting overexpression experiments of Regnase-1 would lend some credibility to the phenotypes, however, this is not pursued in the revised manuscript. Second, while constitutive Regnase-1 knockdown produced robust phenotypes for both sleep-dependent and sleep-independent memory, it also led to a severe short-term memory phenotype. This raises the possibility that flies with constitutive Regnase-1 knockdown are poor learners, thereby having little memory to consolidate. The defect in learning could be simply caused by chronic sleep loss before training. Thus, this set of results does not substantiate a strong link between sleep and long-term memory consolidation. Lastly, the discussion on the sequential function of training, sleep, and RNA processing on memory consolidation appears speculative based on the present data.

We thank the reviewer for the enthusiasm about the approach. As noted above, we have now removed all claims about a link between sleep and memory, and instead just emphasize that we have identified RNA processing genes that affect sleep and memory. We agree with the reviewer that the basis of the Regnase-1 memory phenotype is unclear as the flies may be poor learners. Also, the learning/memory defects could be secondary to sleep loss or, as Reviewer 4 below suggests, all the behavioral deficits could be caused by impaired development/function of the relevant ap ɑ′/β′ cells. We have now included this possibility in the discussion of the manuscript. And we have modified the discussion on training, RNA processing, sleep and memory to emphasize the need for future experiments to address the sequence and relationship of these different processes.

**Reviewer #4 (Public review):**
Summary:Li and Chouhan et al. follow up on a previous publication describing the role of anterior-posterior (ap) and medial (m) ɑ′/β′ Kenyon cells in mediating sleep-dependent and sleep-independent memory consolidation, respectively, based on feeding state in *Drosophila melanogaster*. The authors sequenced bulk RNA of ap ɑ′/β′ Kenyon cells 1h after flies were either trained-fed, trained-starved or untrained-fed and find a small number of genes (59) differentially expressed (3 upregulated, 56 downregulated) between trained-fed and trained-starved conditions. Many of these genes encode proteins involved in the regulation of gene expression. The authors then screened these differentially expressed genes for sleep phenotypes by expressing RNAi hairpins constitutively in ap ɑ′/β′ Kenyon cells and measuring sleep patterns. Two hits were selected for further analysis: Polr1F, which promoted sleep, and Regnase-1, which reduced sleep. The pan-neuronal expression of Polr1F and Regnase-1 RNAi constructs was then temporally restricted to adult flies using the GeneSwitch system. Polr1F sleep phenotypes were still observed, while Regnase-1 sleep phenotypes were not, indicating developmental defects. Appetitive memory was then assessed in flies with constitutive knockdown of Polr1F and Regnase-1 in ap ɑ′/β′ Kenyon cells. Polr1F knockdown did not affect sleep-dependent or sleep-independent memory, while Regnase-1 knockdown disrupted sleep-dependent memory, sleep-independent memory, as well as learning. Polr1F knockdown increased pre-ribosomal RNA transcripts in the brain, as measured by qPCR, in line with its predicted role as part of the RNA polymerase I complex. A puromycin incorporation assay to fluorescently label newly synthesized proteins also indicated higher levels of bulk translation upon Polr1F knockdown. Regnase-1 knockdown did not lead to observable changes in measurements of bulk translation.Strengths:The proposed involvement of RNA processing genes in regulating sleep and memory processes is interesting, and relatively unexplored. The methods are satisfactory.Weaknesses:The main weakness of the paper is in the overinterpretation of their results, particularly relating to the proposed link between sleep and memory consolidation, as stated in the title. Constitutive Polr1F knockdown in ap ɑ′/β′ Kenyon cells had no effect on appetitive long-term memory, while constitutive Regnase-1 knockdown affected both learning and memory. Since the effects of constitutive Regnase-1 knockdown on sleep could be attributed to developmental defects, it is quite plausible that these same developmental defects are what drive the observed learning and memory phenotypes. In this case, an alternative explanation of the authors' findings is that constitutive Regnase-1 knockdown disrupts the entire functioning of ap ɑ′/β′ Kenyon cells, and as a consequence behaviors involving these neurons (i.e. learning, memory and sleep) are disrupted. It will be important to provide further evidence of the function of RNA processing genes in memory in order to substantiate the memory link proposed by the authors.

As noted above, we have removed claims of a link between sleep and memory and instead focused the manuscript on our findings of RNA processing genes modulated during sleep-dependent memory. We concur that impaired development of ap ɑ′/β′neurons could account for the sleep and memory phenotype observed and have included this possibility in the manuscript.

**Recommendations for the authors:**

**Reviewer #4 (Recommendations for the authors):**
The title of the paper should be reconsidered to reflect the results. The evidence for a link between RNA processing genes and memory is weak.

We have changed the title.

Line 328. The term "central dogma" is misused. The central dogma refers to the unidirectional flow of information from DNA to protein. Instead the authors mean "gene expression".

Changed, thank you.

A couple of minor comments relating to the figures:Figure 1b. It is not clear what the number 10570 in the bottom right corner refers to.

Fixed.

Figure 3b. RU- and RU+ annotation is missing (as shown in 3d).

Fixed.